# Application of the Composite Electrical Insulation Layer with a Self-Healing Function Similar to Pine Trees in K-Type Coaxial Thermocouples

**DOI:** 10.3390/s25165210

**Published:** 2025-08-21

**Authors:** Zhenyin Hai, Yue Chen, Zhixuan Su, Hongwei Ji, Yihang Zhang, Shigui Gong, Shanmin Gao, Chenyang Xue, Libo Gao, Zhichun Liu

**Affiliations:** 1School of Aerospace Engineering, Xiamen University, Xiamen 361005, China; haizhenyin@xmu.edu.cn (Z.H.); 35120231151728@stu.xmu.edu.cn (Y.C.); suzhixuan@stu.xmu.edu.cn (Z.S.); 15259300843@163.com (Y.Z.); gaoshanmin@stu.xmu.edu.cn (S.G.); xuechenyang@xmu.edu.cn (C.X.); 2Inner Mongolia Aerospace Power Machinery Testing Institute, Hohhot 010076, China; 17865320564@163.com; 3School of Optoelectronics and Communications Engineering, Xiamen University of Technology, Xiamen 361005, China; 2322101004@stu.xmut.edu.cn; 4Pen-Tung Sah Institute of Micro-Nano Science and Technology, Xiamen 361005, China; lbgao@xmu.edu.cn

**Keywords:** coaxial thermocouple, self-healing, bioinspired, high-temperature insulation, dip-coating

## Abstract

Aerospace engines and hypersonic vehicles, among other high-temperature components, often operate in environments characterized by temperatures exceeding 1000 °C and high-speed airflow impacts, resulting in severe thermal erosion conditions. Coaxial thermocouples (CTs), with their unique self-eroding characteristic, are particularly well suited for use in such extreme environments. However, fabricating high-temperature electrical insulation layers for coaxial thermocouples remains challenging. Inspired by the self-healing mechanism of pine trees, we designed a composite electrical insulation layer with a similar self-healing function. This composite layer exhibits excellent high-temperature insulation properties (insulation resistance of 14.5 kΩ at 1200 °C). Applied as the insulation layer in K-type coaxial thermocouples via dip-coating, the thermocouples were tested for temperature and heat flux. Temperature tests showed an accuracy of 1.72% in the range of 200–1200 °C, a drift rate better than 0.474%/h at 1200 °C, and hysteresis better than 0.246%. The temperature response time was 1.08 ms. Heat flux tests demonstrated a measurable range of 0–41.32 MW/m^2^ with an accuracy better than 6.511% and a heat flux response time of 7.6 ms. In simulated extreme environments, the K-type coaxial thermocouple withstood 70 s of 900 °C flame impact and 50 cycles of high-power laser thermal shock.

## 1. Introduction

Real-time monitoring of surface temperature and heat flux is critical for thermal components in aerospace engines and hypersonic vehicles operating under high-temperature and high-speed conditions [1,2,3,4,5]. Such environments often involve rapid transient changes in temperature and heat flux, demanding high sensor response speeds [6]. Traditional sensors like thin-film thermocouples [7], thin-film thermistors [8,9], and fiber-optic temperature sensors [10,11] are widely used but typically measure only one parameter. They also face limitations in high-temperature-resistance, response speed, and installation conditions [12,13].

Coaxial thermocouples, as self-eroding temperature sensors, have a probe surface that is constantly forming new measurement junctions even as it is constantly being eroded by high-temperature and high-speed airflow. This unique structure enables rapid response, reusability, and adaptability to extreme environments [14,15]. Additionally, their small size allows flexible installation on curved surfaces without disturbing the heat flux field [16]. These advantages make coaxial thermocouples promising for temperature and heat flux measurements in high-temperature and high-speed environments. Anil Kumar Rout et al. used epoxy resin as an insulating layer to manufacture E-type coaxial thermocouples. The temperature and heat flux values measured in continuous wave laser source experiments showed good consistency in terms of trend and amplitude, with an uncertainty range of ±5% [17]. However, most existing studies use epoxy resin as insulation material, which cannot withstand temperatures above 400 °C. This limits the thermocouple’s measurement range. Byrenn Birch et al. proposed alumina as the insulation layer for E-type coaxial thermocouples, opening new avenues for insulation materials. In a low-enthalpy hypersonic wind tunnel operating at 6 Mach, the team successfully measured instantaneous heat flux density values of approximately 10 kW/m^2^ at the stagnation point [18]. Sanjeev Kumar Manjhi et al. used Teflon to form a 10–20 μm insulating layer. Using a self-made shock tube flow facility, under high-pressure shock action for 60 milliseconds, the transient temperature and surface heat flux errors of the coaxial thermocouple in four experiments were less than ±0.2% and ±4%, respectively [19]. Chen Jun et al. used high-temperature adhesives as insulation layers for tungsten–rhenium coaxial thermocouples, achieving a measurable heat flux of up to 21 MW/m^2^ [20]. Therefore, improving the insulation layer’s performance is key to enhancing the K-type coaxial thermocouple’s temperature and heat flux measurement capabilities.

This paper reports on the application of a high-temperature-resistant, highly insulating composite electrical insulation layer in K-type coaxial thermocouples. Inspired by the self-healing mechanism of pine trees—where damaged bark secretes resin to seal wounds and solidify into a protective layer—we designed a composite electrical insulation layer composed of silicate and alumina. The layer was applied to K-type coaxial thermocouples and the temperature and heat flux were tested. Experimental results demonstrated excellent performance, providing a viable solution for accurate temperature and heat flux data acquisition in extreme environments.

## 2. Materials and Methods

### 2.1. Materials

Ni-Cr10 tubes, Ni-Cr10 wires, and Ni-Si3 wires are produced by Yuguanghong Trading Co., Ltd. (Zhenzhou, China). The Ni-Cr10 tubes have an outer diameter of 2 mm, an inner diameter of 0.52 mm, and a length of 20 mm. Both the Ni-Cr10 wires and Ni-Si3 wires have a diameter of 0.5 mm. Silicate solution used to form the silicate electrical insulation layer was purchased from Pfizer Electronics Co., Ltd. (07H-3509, Dongguan, China). Alumina solution preparation materials used to form alumina electrical insulation layers included nano alumina sol, alumina powder, and alumina fiber wool. Nano alumina sol was purchased from Jinghuo Technical Glass Co., Ltd. (Dezhou, China). Alumina powder (10 nm) was purchased from Yaoyi Alloy Co., Ltd. (Shanghai, China). Alumina fiber wool was purchased from Rui Zhi Youth Network Technology Co., Ltd. (Pingdingshan, China). A B-type thermocouple was produced by Xinghua Shunsheng Electric Co., Ltd. (Hefei, China), with a tolerance of ±1.5 °C. A standard heat flux sensor was purchased from Thermal Characteristics Sensing Technology Co, Ltd. (CTK-2-1M, Shanghai, China).

### 2.2. Processing, Testing and Characterization Equipment

Dip-coating machine (DR-TP600, Guangdong South China Technology Co., Ltd., Guangzhou, China) was used to apply a silicate electrical insulation layer and an alumina electrical insulation layer to the surface of Ni-Si3 wire. Flash Joule heating furnace (FHT3200-2025Q1, Xiamen Guangrui Sensing Technology Co., Ltd., Xiamen, China) was used for rapid pre-sintering of CTs. Tube furnace (OTF-1200X, Hefei Kexing Materials Technology Co., Ltd., Hefei, China) was used for sintering CTs and providing the heat source for the static calibration test section. Laser welding machine (AHL-W600III, Shenzhen Aohua Laser Technology Co., Ltd., Shenzhen, China) was used to weld Ni-Cr10 alloy wires onto Ni-Cr10 tubes and provide the heat source for the heat flux test section. A digital multimeter (DAQ6510, TEKTRONIX, INC., Beaverton, OR, USA) and a miniature dynamic data acquisition and analysis system (DH5916, Jiangsu Donghua Testing Technology Co., Ltd., Taizhou, China) were used to measure the insulation resistance of the composite electrical insulation layer and the thermo-electric potential of CTs. Scanning electron microscopy (SUPRA55 SAPPHIRE, Carl Zeiss AG, Oberkochen, Germany) was used to capture microscopic morphology images of the composite electrical insulation layer.

### 2.3. Preparation of K-Type Coaxial Thermocouple

The K-type coaxial thermocouple consists of a Ni-Cr10 tube, Ni-Cr10 wire, Ni-Si3 wire, and a composite electrical insulation layer. Figure 1a shows the overall structural diagram of the K-type coaxial thermocouple, and Figure 1b shows the exploded view of the K-type coaxial thermocouple. The composite electrical insulation layer includes a silicate base layer and an alumina top layer. Silicate was chosen as the base layer because of its excellent adhesion to nickel [21]. The dip-coating method was used for its cost-effectiveness and suitability for curved surfaces [22]. Figure 1c shows the silicate solution used to form the silicate electrical insulation layer. The alumina solution was prepared by stirring alumina sol, alumina powder, and alumina fiber powder in a weight ratio of 10:1:2 and stirring for 6 h (as shown in Figure 1d). Alumina fiber powder was prepared by grinding alumina fiber wool into powder using a mortar.

Figure 1e shows the manufacturing process for the K-type coaxial thermocouple. Prior to fabrication, Ni-Cr10 tubes and Ni-Si3 wires underwent ultrasonic cleaning sequentially with acetone, deionized water, and alcohol to remove surface impurities and oxides. The first step involved dip-coating Ni-Si3 wires with silicate solution using a dip-coating machine, with parameters set to an immersion time of 60 s, withdrawal speed of 20 μm/s, and immersion depth of 50 mm. The second step involved heating the silicate-coated wires with a hot-air gun at 200 °C for 10 s to form a silicate electrical insulation layer, followed immediately by rapid pre-sintering in a flash Joule heating furnace at 1100 °C for 2 min. The third step involved dip-coating the wires with alumina solution under identical parameters (60 s immersion, 20 μm/s withdrawal, 50 mm depth). The fourth step involved injecting alumina solution into the Ni-Cr10 tube using a syringe and inserting the Ni-Si3 wire. The fifth step involved drying the assembly in a 30 °C oven for 2 h to solidify the alumina solution into an alumina electrical insulation layer, thereby forming the composite electrical insulation layer. The sixth step involved sintering the assembly in a tube furnace at 900 °C for 30 min. The seventh step involved welding the Ni-Cr10 tube to the Ni-Cr10 wire. Finally, the measurement junction was formed by grinding the probe surface with sandpaper. Figure 2a shows an image of the probe surface of the K-type coaxial thermocouple, and Figure 2b shows a cross-sectional view of a K-type coaxial thermocouple.

### 2.4. Performance Parameter Evaluation Method

In this study, the performance parameters of K-type coaxial thermocouples, including hysteresis, stability/drift, accuracy, and temperature and heat flux response time, are investigated.

Hysteresis refers to the deviation in measurement results at the same temperature point during the temperature rise and fall process of the thermocouple being measured. Hysteresis is calculated by the following formula [23]:(1)$$\mathrm{Hysteresis}=\frac{{\left.\Delta V\right|}_{T=x}}{V_{F.S.}}\times 100\%$$
where ${\left.\Delta V\right|}_{T=x}$ denotes the deviation of the K-type coaxial thermocouple output voltage value at the same temperature point during the temperature rise and fall process, and $V_{F.S.}$. denotes the voltage value corresponding to the calibration temperature range.

Stability/drift represents the rate at which the output temperature signal of the thermocouple being measured changes unexpectedly over time when operating at a constant temperature. Stability/drift is calculated as follows [24]:(2)$$\mathrm{Drift}/\mathrm{Stability}=\frac{\Delta T_{\mathrm{CT}}-\Delta T_{B}}{T_{\mathrm{set}}\times t}\times 100\%$$
where t denotes the test duration. $\Delta T_{\mathrm{CT}}$ denotes the maximum change in the temperature reading of the K-type coaxial thermocouple during the test duration. $\Delta T_{B}$ denotes the maximum change in the temperature reading of the standard B-type thermocouple during the test duration. $T_{\mathrm{set}}$ denotes constant temperature set during the test duration.

Accuracy refers to the maximum permissible deviation between the measured value of the thermocouple being measured and the true value. Accuracy is calculated by the following formula [25]:(3)$$\mathrm{Accuracy}=\frac{\Delta T}{T_{F.S.}}\times 100\%$$
where ΔT denotes the temperature deviation between the K-type coaxial thermocouple and the standard B-type thermocouple, and $T_{F.S.}$ denotes the calibration temperature range.

Response time ($\tau_{0.632}$) refers to the time required for the thermocouple output temperature or heat flux value to track the step change in the measured temperature or heat flux value. Response time is calculated using the following formula [26]:(4)$$\tau_{0.632}=\Delta t\times 63.2\%$$
where Δt denotes the time for the thermocouple to reach a steady state after being excited by a heat source.

The thermal product coefficients calibration of the thermocouple being measured is calculated using the following formula [20]:(5)$${\sqrt{\rho \mathrm{ck}}}_{\mathrm{CT}}={\sqrt{\rho \mathrm{ck}}}_{\mathrm{glycerol}}\times \left(\frac{T_{10}-T_{20}}{T_{e}-T_{20}}-1\right)$$
where ${\sqrt{\rho \mathrm{ck}}}_{\mathrm{CT}}$ denotes the thermal product coefficients of the thermocouple being measured. ${\sqrt{\rho \mathrm{ck}}}_{\mathrm{glycerol}}$ denotes the thermal product coefficients of glycerol, $T_{10}$ denotes the temperature of glycerol, $T_{20}$ denotes the temperature of the thermocouple being measured before it is immersed in glycerol, and $T_{e}$ denotes the equilibrium temperature of the transient at the interface after the thermocouple comes into contact with glycerol.

Based on the one-dimensional semi-infinite body assumption [16], the temperature data collected by the thermocouple being measured is converted to the specific surface heat flux rate using the Cook–Felderman method [27]. The specific surface heat flux rate is calculated according to the following equation [15]:(6)$$q\left(t_{n}\right)=\frac{2\sqrt{\rho \mathrm{ck}}}{\alpha \sqrt{\pi }}\overset{n}{\underset{j=1}{\sum }}\frac{T_{j}-T_{j-1}}{\sqrt{t_{n}-t_{i}}+\sqrt{t_{n}-t_{i-1}}}$$
where $q\left(t_{n}\right)$ denotes heat flux at the moment $t_{n}$ (W/m^2^), ρ is density (g/m^3^), c is specific heat capacity (J/g·°C), k is thermal conductivity (W/m·°C), α denotes calibration factor, $T_{j}$ and $T_{j-1}$ denotes temperature at moments tj and tj−1 (°C).

Heat flux error is calculated by the following formula:(7)$$\mathrm{Heat}\;\mathrm{flux}\;\mathrm{error}=\frac{\left|\Delta q\right|}{q_{\mathrm{Standard}\;}}\times 100\%$$
where Δq denotes the heat flux deviation between the thermocouple being measured and the standard heat flux sensor, and $q_{\mathrm{Standard}\;}$ denotes the heat flux measured by standard heat flux sensors.

## 3. Results and Discussion

The performance of the electrical insulation layer in K-type coaxial thermocouples directly determines temperature and heat flux measurement accuracy. Previous studies indicate that when pine bark is damaged, pine resin exudes to cover the wound and solidifies into a protective layer (as shown in Figure 3a), preventing pathogen invasion through a self-healing mechanism [28]. Inspired by this, we designed a composite electrical insulation layer based on the principle of “combining soft and hard materials” [29]. The composite electrical insulating layer uses high-melting-point alumina ceramic material as the base layer, serving as the “hard” component, providing structural support and thermal stability for the composite electrical insulating layer. The silicate glass ceramic material acts as the “soft” top layer, melting and flowing at high temperatures to fill cracks.

Figure 3b illustrates the microstructural evolution of the composite electrical insulating layer at different sintering temperatures. Between 800 °C and 1200 °C, thermal stress causes cracks to form in the alumina oxide insulating layer. These cracks provide low-resistance pathways, significantly increasing leakage currents at high temperature and accelerating insulation failure. When the sintering temperature reaches 1200 °C, the silicate electrical insulation layer undergoes a melting phase transformation [21,30]. The viscous glass phase dynamically penetrates along the cracks in the alumina electrical insulation layer, autonomously filling micro-cracks to achieve self-healing (as shown in Figure 3c). This process simulates the flow sealing mechanism of pine resin. EDS surface mapping confirms Si enrichment in the crack self-healing regions and Al/O dominance elsewhere, further validating the self-healing function. Figure 3d shows EDS spectra with weight percentages: O (54.41%), Si (13.99%), Al (28.62%), Mg (2.29%), and Ca (0.69%).

To further investigate the enhancement of electrical insulation performance via the self-healing mechanism of the composite electrical insulation layer, insulation resistance tests were conducted on K-type coaxial thermocouples individually coated with the silicate electrical insulation layer, alumina electrical insulation layer, and composite electrical insulation layer. The fixed dimensions of the Ni-Cr10 tube (inner diameter 0.52 mm) and Ni-Si3 wire (diameter 0.5 mm) ensure a consistent insulation layer thickness of 20 μm for all samples. Results (Figure 3e) showed that the alumina-only layer achieved only 2 kΩ at 1050 °C, while the silicate-only layer reached 3.3 kΩ at 1190 °C. In contrast, the composite electrical insulation layer exhibited the significantly higher resistance of 14.5 kΩ at 1200 °C. This demonstrates a substantial improvement in electrical insulation performance compared to the single-component layers.

Static temperature calibration testing is primarily used to evaluate the temperature measurement accuracy and stability of the K-type thermocouple under different temperature conditions. Figure 4 shows the overall schematic diagram of the static temperature calibration platform, which consists of a tubular furnace, a standard B-type thermocouple, compensating conductors, an ice–water mixture, copper wires, a digital multimeter, and a computer. The temperature measurement probes of the K-type thermocouple and standard B-type thermocouple were placed together in the constant-temperature zone of the tubular furnace. The cold-end of both was extended to the ice–water mixture via compensating conductors to eliminate errors caused by temperature gradients at the cold-end of the thermocouples.

Figure 5a shows the K-type coaxial thermocouple voltage–temperature conversion relationship and fitting curve obtained by three polynomial fittings (R^2^ = 0.9998). The fitting error remained below 0.23% (as shown in Figure 5b). Accuracy evaluation showed a deviation of 1.72% compared to the standard B-type thermocouple (as shown in Figure 5c). The three-cycle heating and cooling test (Figure 5d) confirmed stable operation for over 12 h. The drift rate was 0.474%/h at 1200 °C (as shown in Figure 5e). Hysteresis remained below 0.246% (as shown in Figure 5f). The K-type coaxial thermocouple showed excellent temperature measurement performance in static calibration test, which further proved the insulation reliability of the composite electrical insulation layer with the self-healing function at high temperature.

To evaluate the K-type coaxial thermocouple’s dynamic response to thermal excitation, temperature response time tests were performed. The test platform (Figure 6a) included a digital multimeter, computer, laser generator, stainless fixture, and ice–water mixture. The thermocouple was mounted on the stainless fixture with its probe surface flush with the fixture surface (as shown in Figure 6b). A stable laser-generated heat source was applied. Results (Figure 6c) showed a temperature response time of 1.55 ms, demonstrating rapid temperature tracking capability.

Before heat flux testing, the thermal product coefficients of the K-type coaxial thermocouple were calibrated using the sudden-immersion method [31] to ensure measurement accuracy. As shown in Figure 7, the calibration platform comprised a digital multimeter, computer, fixed bracket, glycerol, platinum resistance, adhesive tape, and heating stage. A beaker containing glycerol was placed on the heating stage set to a constant 150 °C. The platinum resistance was fixed to the bracket with adhesive tape and immersed in glycerol to monitor its temperature. Glycerol has a thermal product coefficient of 939 WS^1/2^/(m^2^K) [20]. The thermocouple was hand-immersed into glycerol, and the initial temperature before immersion and the transient equilibrium temperature at the glycerol interface were recorded. Equation (5) calculates the thermocouple’s thermal product coefficients. Figure 8a shows typical calibration curves. Six calibrations yielded values ranging from 7683 to 8936 WS^1/2^/(m^2^K) (as shown in Figure 8b). The variation likely stems from manual immersion control during calibration. To mitigate this, the average value (8486 WS^1/2^/(m^2^K)) was adopted and corrected using calibration factor α in Equation (6). The thermal product coefficient of the standard heat flux sensor was 8900 WS^1/2^/(m^2^K) [19,32].

The heat flux test platform was the same as the temperature response time test platform (as shown in Figure 6a). The sensor was mounted on the stainless fixture, its surface flush with the fixture. Adjust the laser spot parameter to −9 to ensure that the laser spot size fully covers the sensor probe surface. To further ensure that the sensor absorbs the heat flux generated by the laser, a high-temperature absorption coating was coated on the surface of the sensor probe [33]. Four laser levels (40 V, 50 V, 60 V, and 70 V) were used. The heat flux density of the standard heat flux sensor under four laser levels was 17.83 MW/m^2^, 22.35 MW/m^2^, 35.48 MW/m^2^, and 43.35 MW/m^2^, respectively (as shown in Figure 8c). Repeating the test with the K-type coaxial thermocouple yielded 18.65 MW/m^2^, 21.83 MW/m^2^, 33.17 MW/m^2^, and 41.32 MW/m^2^ (as shown in Figure 8d). Figure 8e shows the heat flux error of K-type coaxial thermocouple under four laser levels, and the maximum error was 6.511%. The heat flux response time was 7.6 ms (as shown in Figure 8f). These results confirm the ability of K-type coaxial thermocouples to measure ultra-high heat flux in excess of 40 MW/m^2^, with an excellent heat flux measurement range.

To evaluate the temperature and heat flux measurement capabilities of K-type coaxial thermocouples in extreme environments, high-temperature flame impact and high-power laser cyclic thermal shock tests were conducted. The high-temperature flame impact was used to simulate a continuous thermal load environment, with the high-temperature flame from a butane flame gun directed at the sensing probe surface of the K-type coaxial thermocouple (as shown in Figure 9a). The K-type coaxial thermocouple remained undamaged during a flame thermal shock test lasting up to 70 s at 900 °C with a maximum heat flux of 2.2 MW/m^2^ (as shown in Figure 9b). The high-power laser cyclic thermal shock test was used to simulate a rapid thermal cycling environment. After 50 cycles of high-power laser thermal impact at temperatures up to 1000 °C, the K-type coaxial thermocouple maintained stable response characteristics (as shown in Figure 9c). The experimental data clearly demonstrated that the K-type coaxial thermocouple can effectively measure temperature and heat flux data under extreme operating conditions.

## 4. Conclusions

Inspired by the self-healing mechanism of pine trees, we designed a composite electrical insulation layer composed of a silicate base layer and an alumina top layer. The composite electrical insulation layer self-heals micro-cracks at high temperatures, achieving excellent insulation performance (14.5 kΩ at 1200 °C). Applied to the K-type coaxial thermocouple via dip-coating, the thermocouple was tested for temperature and heat flux. Temperature tests demonstrated an accuracy of 1.72% from 200 °C to 1200 °C, a drift rate better than 0.474%/h at 1200 °C, and hysteresis better than 0.246%. The temperature response time was 1.08 ms. Heat flux tests reveal a measurable range of 0–41.32 MW/m^2^ with accuracy better than 6.511% and a heat flux response time of 7.6 ms. Finally, the K-type coaxial thermocouple was applied in 70 s of 900 °C flame impact tests and 50 high-power laser cyclic thermal shock tests, simulating the operating conditions of aerospace engine thermal components. The K-type coaxial thermocouple demonstrated excellent temperature and heat flux measurement performance in the simulated tests, indicating that it provides a feasible solution for obtaining temperature and heat flux data of aerospace engine thermal components.

## Figures and Tables

**Figure 1 sensors-25-05210-f001:**
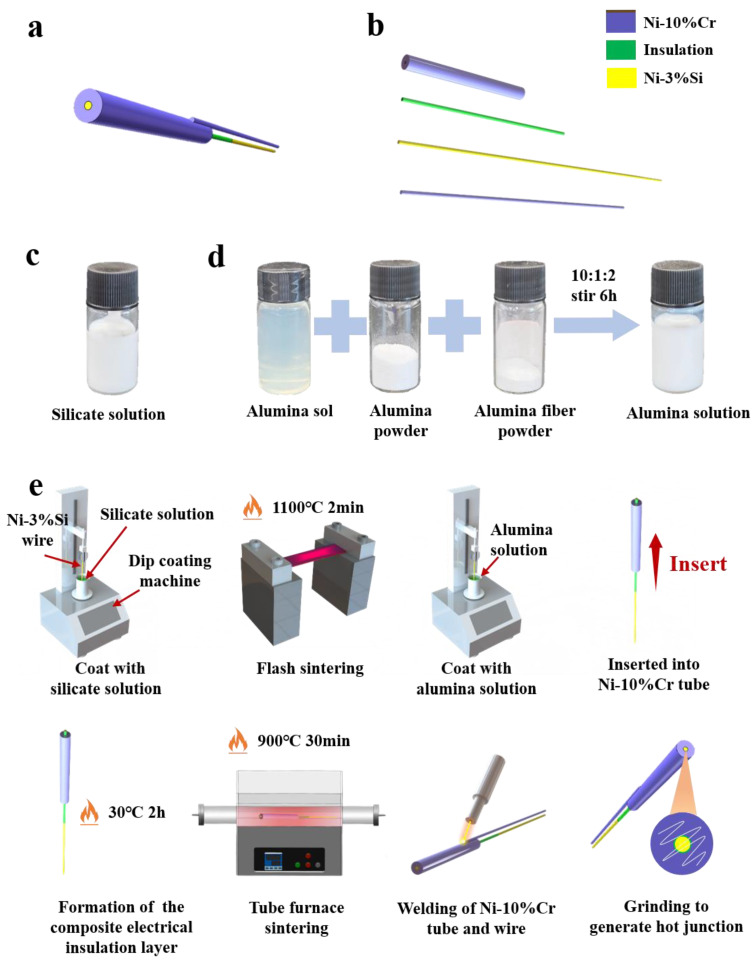
(**a**) The overall structural diagram of the K-type coaxial thermocouple; (**b**) the exploded view of the K-type coaxial thermocouple; (**c**) actual image of silicate solution; (**d**) schematic diagram of alumina solution preparation; (**e**) the manufacturing process for the K-type coaxial thermocouple.

**Figure 2 sensors-25-05210-f002:**
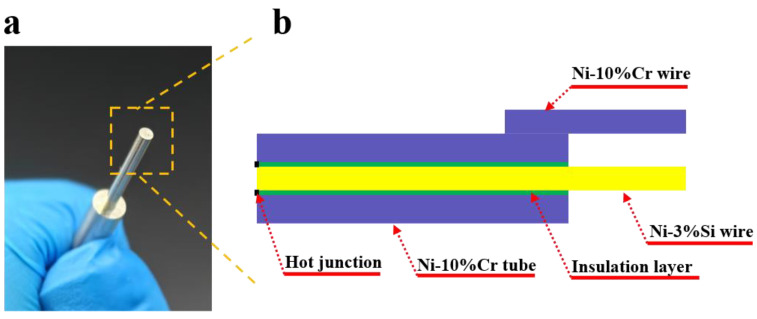
(**a**) The image of the probe surface of the K-type coaxial thermocouple; (**b**) the exploded view of the K-type coaxial thermocouple.

**Figure 3 sensors-25-05210-f003:**
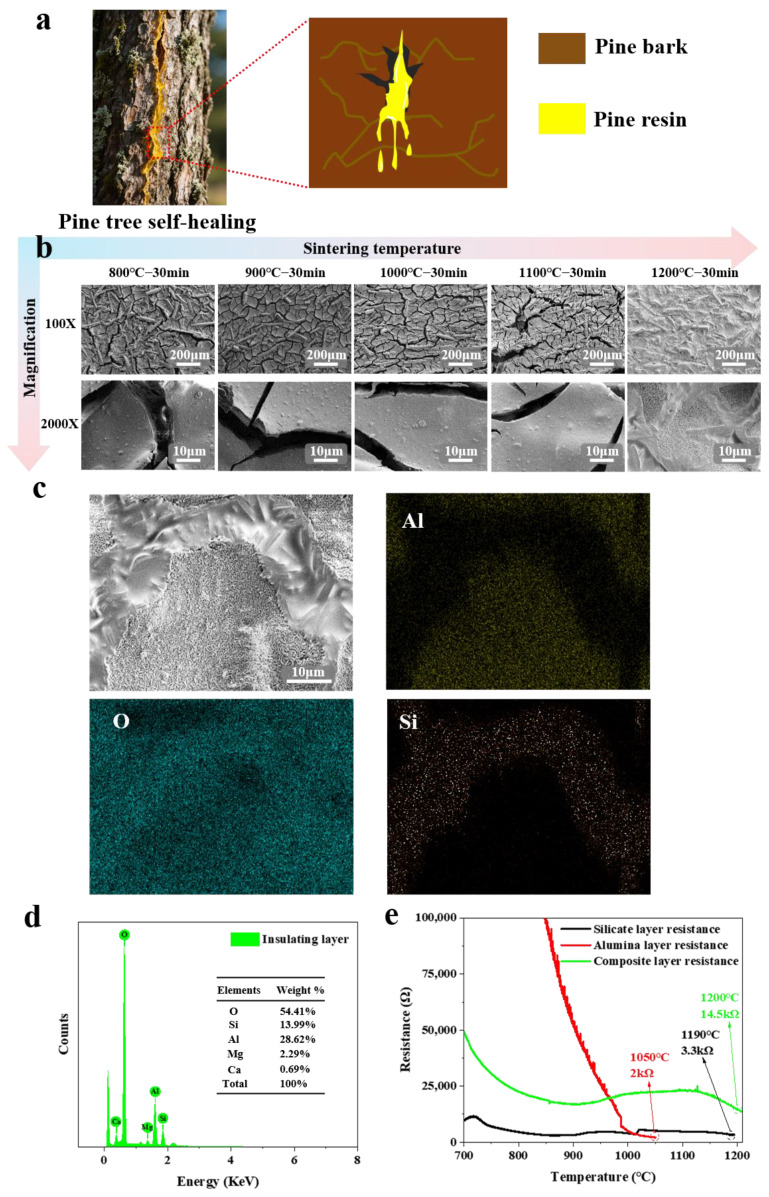
(**a**) Schematic diagram of the self-healing wound of pine secretory resin; (**b**) microstructure evolution of the composite electrical insulation layer at different sintering temperatures; (**c**) EDS surface distribution of the composite electrical insulation layer; (**d**) EDS spectra of the composite electrical insulation layer; (**e**) comparison of insulation resistance.

**Figure 4 sensors-25-05210-f004:**
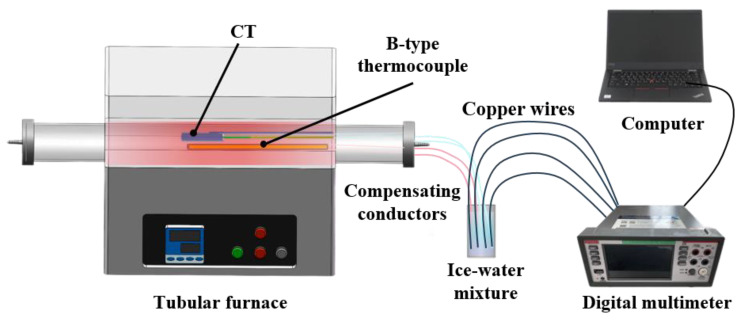
Static calibration testing platform for the K-type coaxial thermocouple.

**Figure 5 sensors-25-05210-f005:**
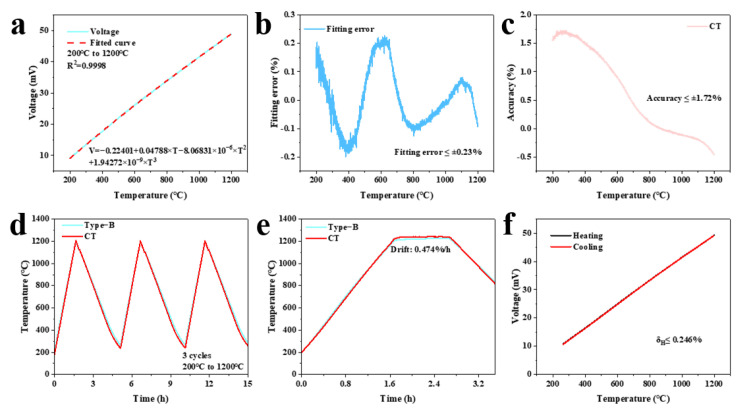
(**a**) Voltage curves and fitting curve of the K-type coaxial thermocouple; (**b**) fitting error curve of the K-type coaxial thermocouple; (**c**) accuracy curve of the K-type coaxial thermocouple; (**d**) three-cycle heating and cooling test results of K-type coaxial thermocouple; (**e**) drift rate test results of the K-type coaxial thermocouple; (**f**) hysteresis test results of the K-type coaxial thermocouple.

**Figure 6 sensors-25-05210-f006:**
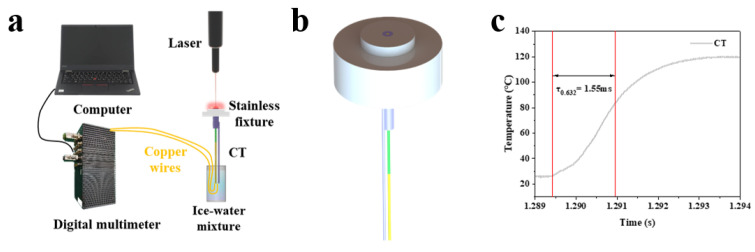
(**a**) Temperature response time and heat flux testing platform; (**b**) schematic diagram of stainless fixture with K-type coaxial thermocouple; (**c**) response time test results of the K-type coaxial thermocouple.

**Figure 7 sensors-25-05210-f007:**
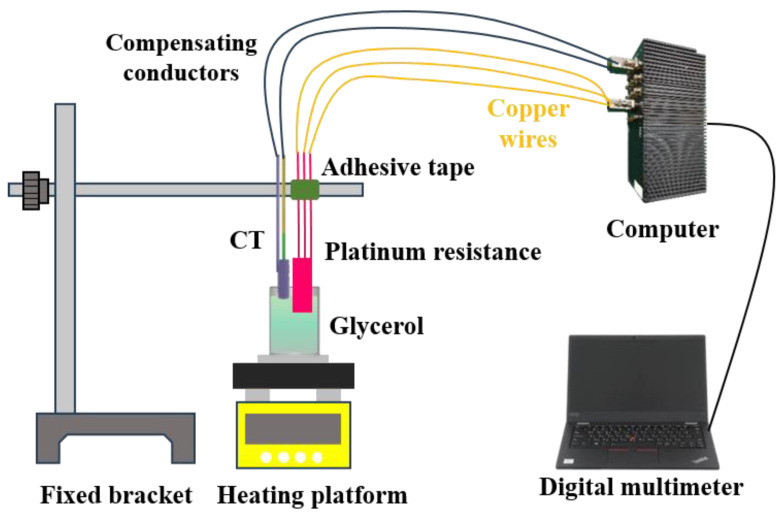
K-type coaxial thermocouple thermal product coefficients calibration platform.

**Figure 8 sensors-25-05210-f008:**
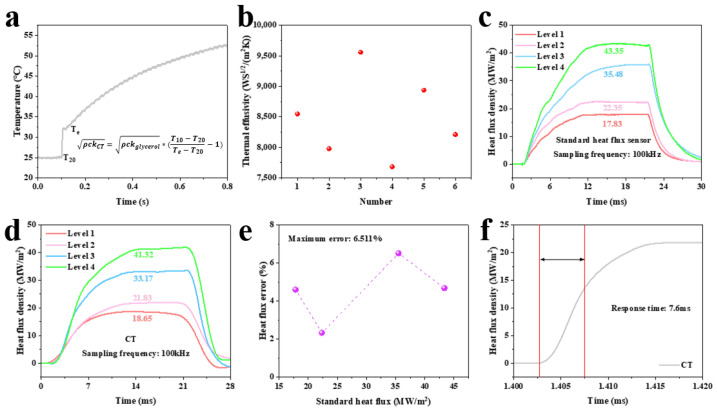
(**a**) Typical curves for calibration of the thermal product coefficients of K-type coaxial thermocouples; (**b**) the thermal product coefficients of K-type coaxial thermocouple obtained by six calibrations; (**c**) test results of heat flux density of a standard heat flux sensor under four kinds of laser levels; (**d**) test results of heat flux density of a K-type coaxial thermocouple under four kinds of laser levels; (**e**) heat flux error of a K-type coaxial thermocouple under four kinds of laser levels; (**f**) heat flux response time of a K-type coaxial thermocouple.

**Figure 9 sensors-25-05210-f009:**
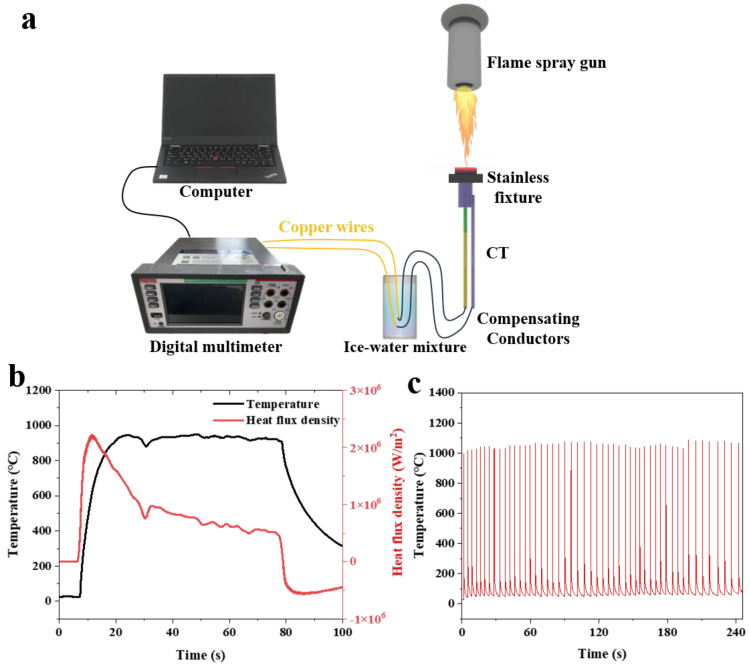
(**a**) High-temperature flame impact testing platform; (**b**) flame impact test results; (**c**) high-power laser cyclic thermal shock test results.

## Data Availability

The original contributions presented in this study are included in the article. Further inquiries can be directed to the author.

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
