# Peer review of "Application of the Composite Electrical Insulation Layer with a Self-Healing Function Similar to Pine Trees in K-Type Coaxial Thermocouples"

_sensors, 2025, doi:10.3390/s25165210_

Round 1
Reviewer 1 Report
Comments and Suggestions for Authors
Dear Authors, let me copy my comments on the very interesting paper: Application of the Composite Electrical Insulation Layer with Self-Healing Function Similar to Pine Trees in K-Type Coaxial Thermocouple.
1. Please introduce in the abstract the practical application of high-temp insulation a bit more. (What is extremely high-temp, and high-speed?)
2. Discussion of self-healing mechanisms would need much deeper definition and discussion. It is not discussed in scientific manner. Altogether, overall literature survey is thin, i'd recommend aspects of the same information requested in comment 1. The mention of "pine trees" is also a bit unnecessary in the title, as there is no deep discussion on this aspect of the paper.
3. Please show the used materials in images, at least on schematic ones. A mind map of experiments would be welcome.
4. Figure 1 is interesting, but i recommend sans-serif fonts for better readability along the paper (later plots suffer from difficult reading). Also it is a bit much to use a-b-c in Figure 1, then i-viii on the same figure. Later on please omit colored backgrounds from plots, it causes distraction.
5. What is the actual meaning of the polynomial fitting?
6. Please define response time (time constant?). Please define it and all the fitting, and calculation/signal processing aspects in the materials and methods section.
Reviewer 2 Report
Comments and Suggestions for Authors
In the article the author uses Ni-Cr10 tubes and Ni-Si3 wires to construct a coaxial thermocouple,and systematically analyzed the static and dynamic output characteristics of the thermocouple.
- In Figure 4, the static output characteristics of the thermocouple are characterized. The thermocouple is placed as a whole in a high-temperature furnace and led out through wires. According to the basic principle of the hot spot thermocouple, the temperature difference between the inside and outside of the furnace is mainly borne by the wires. The thermoelectric potential generated by the coaxial sensor is worth confirming again. This static calibration method should be improved.
- What is the connection method between coaxial sensors and wires at high temperatures, such as 1200 oC, and how to ensure their normal connection? What are its contact characteristics?
- For the introduction section, it is recommended to provide a more detailed and systematic description of the latest developments in coaxial temperature sensors or NiGr/NiSi thermocouples. Suggest a more in-depth analysis of its characteristics. In addition, the output value of this coaxial thermocouple is significantly different from that of the standard K-type thermocouple, and it should be classified as a non-standard thermocouple. Therefore, its definition as K-type should be further determined.
4. Please carefully check the physical quantity format, graphic description, and horizontal and vertical coordinates in the manuscript. For example, keyword capitalization.
Comments on the Quality of English LanguagePlease carefully check the physical quantity format, graphic description, and horizontal and vertical coordinates in the manuscript. For example, keyword capitalization.
Reviewer 3 Report
Comments and Suggestions for Authors
The manuscript by Zhenyin HAI et al. presents the application of a high-temperature-resistant, highly insulating composite electrical insulation layer composed of silicate and alumina in a K-type coaxial thermocouple. Overall, the manuscript has potential for publication after the following comments are adequately addressed:
- It would be beneficial if the authors could present a clearer schematic illustrating the entire fabrication process of the insulation layer. Additionally, the components shown in Figure 1C should be clearly labeled and explained.
- The equations on page 5 should be properly cited and referenced within the text for clarity and traceability.
- The mechanism and investigation regarding the self-healing behavior of the insulation layer should be more clearly explained and discussed in the manuscript.
Round 2
Reviewer 1 Report
Comments and Suggestions for Authors
Dear Authors, i recommend one last thing. To remove colored backgrounds from figures. It is really distracting and they are not in line with the standards of publication.
The rest of the comments were addressed fine, thank you!
